# Magnetically Modified Biosorbent for Rapid Beryllium Elimination from the Aqueous Environment

**DOI:** 10.3390/ma14216610

**Published:** 2021-11-03

**Authors:** Michaela Tokarčíková, Oldřich Motyka, Pavlína Peikertová, Roman Gabor, Jana Seidlerová

**Affiliations:** Nanotechnology Centre, Energy and Environmental Technology Centre, VŠB-Technical University of Ostrava, 708 33 Ostrava-Poruba, Czech Republic; oldrich.motyka@vsb.cz (O.M.); pavlina.peikertova@vsb.cz (P.P.); roman.gabor@vsb.cz (R.G.); jana.seidlerova@vsb.cz (J.S.)

**Keywords:** magnetic biochar, different Fe content, beryllium, non-linear sorption, elimination, kinetics

## Abstract

Although both beryllium and its compounds display high toxicity, little attention has been focused on the removal of beryllium from wastewaters. In this research, magnetically modified biochar obtained from poor-quality wheat with two distinct Fe_x_O_y_ contents was studied as a sorbent for the elimination of beryllium from an aqueous solution. The determined elimination efficiency was higher than 80% in both prepared composites, and the presence of Fe_x_O_y_ did not affect the sorption properties. The experimental *q_max_* values were determined to be 1.44 mg/g for original biochar and biochar with lower content of iron and 1.45 mg/g for the biochar with higher iron content. The optimum pH values favorable for sorption were determined to be 6. After the sorption procedure, the sorbent was still magnetically active enough to be removed from the solution by a magnet. Using magnetically modified sorbents proved to be an easy to apply, low-cost, and effective technique.

## 1. Introduction

Potentially toxic elements (PTEs) naturally occur in minerals deposited in rocks or minerals in nature. However, due to global industry and the persistence of metals that cannot be degraded, heavy metals are accumulated through the food chain in surface water, groundwater, or soils [1,2]. Beryllium occurs naturally in soils at a concentration range less than 15 mg/kg, in low concentration in natural surface waters [3], and it is generally found in plant samples at low concentrations (<1 mg/kg dry weight), as well as in various fish and other marine organisms (<0.1 mg/kg fresh weight) [4].

Although both beryllium and beryllium compounds are classified as carcinogens, beryllium is an important metal in industry, where it is widely used in aircraft and space vehicle construction, instruments, X-ray machines, mirrors, as well as in nuclear weapons and reactors [5]. Smelting and mining of ores lead to beryllium release into industrial wastewaters, and beryllium emissions result from mining-related environmental pollution [6]. The primary route of beryllium exposure to the human body is the inhalation of dust and fumes [4], which may induce lung cancer [7], berylliosis, or pulmonary granulomatosis [8]. Due to beryllium accumulation in the environment, its concentration increases which poses a threat for both the environment and living organisms [9,10], especially at low pH conditions [11,12]. Based on the results of Jagoe et al. [13], beryllium has similar toxic properties as aluminum in acidic water; however, beryllium is more toxic even at low concentrations, e.g., 10 µg/L [12].

Due to the widespread use of beryllium and its known negative impact on the health and environment, it is crucial to eliminate the potential risks. Studies focused on the elimination of beryllium ions from aqueous environments have been sparse [14,15,16,17].

There are several methods of heavy metals extraction or elimination procedures, such as, e.g., membrane filtration [18], solvent extraction [19], chemical precipitation [20], ion exchange, and electrodialysis [21]. Sorption is considered to be superior to all the above methods due to its simplicity, low impact on the environment, and cost-effectiveness [22]. Sorption-based procedures do not produce significant amounts of by-products and, moreover, the sorbent can be recycled and repeatedly used without a significant decrease of sorption capacity [23,24].

Magnetic composites seem to be efficient materials for eliminating pollutants from the aqueous environment due to their high sorption capacity and easy removal by magnetic field post-sorption. Many non-magnetic materials, e.g., clays [25], chitosan/graphene oxide [26], carbon-matrix composite [27], polymer composite [28], biochar [29], or peanut husks [30] can be magnetised. Biochar is a carbon-rich material obtained from the thermochemical conversion of biomass in a closed container with little or no available oxygen [31,32]. Waste biomass is an interesting, sustainable, and promising feedstock for long-term biochar preparation [31]. The feedstock used for biochar preparation can be an organic material such as various wood-based products [33], crop residue, or sewage sludge [34]. Although individual biochars do not have the same properties, due to the different feedstock used and various pyrolysis conditions applied [35], biochar can be generally characterized by porous structure, large specific surface area, abundant mineral components, surface functional groups, and promising sorption properties [36].

Several methods of preparation of magnetic biochar exist. The impregnation-pyrolysis method enables pyrolysis and magnetization in one step and facilitates the control of the physicochemical properties and sorption capacity of the magnetic biochar by controlling the operating parameters [37]. The co-precipitation method is based on mixing the biochar with a solution containing Fe^3+^/Fe^2+^ oxides and altering the pH 10~11 (by adding hydroxide) under stirring [38]. During the solvothermal process, the biochar is added to the mixture containing an iron source, stirred to obtain a precursor mixture, and then heated in an autoclave [39]. Magnetic biochar can be prepared by pyrolysis of pretreated biomass in downstream nitrogen and under defined temperature conditions [40]. For the magnetization of non-magnetic materials, a rapid and simple method has been developed by Safarik and Safarikova [41]. The magnetic iron microparticles prepared using a microwave-assisted procedure were mixed with biochar [41,42].

The material tested in the present paper was obtained from poor-quality wheat (due to drought or high content of ergot), which cannot be processed or otherwise utilized. Poor-quality wheat, together with by-products of wheat processing originating from local sources, seems to be a permanent source for biochar preparation in the future.

## 2. Materials and Methods

### 2.1. Materials

The waste material was obtained from poor-quality wheat (WB) originating from the Czech Republic. Raw organic material was pyrolyzed at 600–640 °C for 3.5 h using the pyrolysis unit PYROMATIC built by Arrow Line in Ostrava Vítkovice (Ostrava Vítkovice, Czech Republic). The product was milled using the planetary micro mill Pulverisette 7 FRITSCH 10. FeSO_4_·7H_2_O (purum p.a., MACH CHEMIKÁLIE s.r.o., Ostrava, Czech Republic) was used as a precursor for Fe_x_O_y_ NPs. NaOH (purum p.a., MACH CHEMIKÁLIE s.r.o., Ostrava, Czech Republic) was used to prepare 1 mol/L solution needed for pH adjustment during composites preparation.

### 2.2. Preparation of Magnetic Sorbents

Two magnetic waste biochars (MWB) with different Fe amounts were prepared using the two-step method. A total of 0.8 g and 1.4 g of FeSO_4_·7H_2_O were separately dissolved in 150 mL of deionized water, then NaOH solution (1 mol/L) was slowly added. The mixtures were continuously stirred to precipitate the iron hydroxide. After reaching 11–12 pH, the suspensions were diluted to 250 mL with water and placed into a standard microwave oven (700 W, 2450 MHz) for 10 min. Then, the SO_4_^2−^ ions were eliminated by decantation (evidence reaction using BaCl_2_). In the second step, about 10 mL of the suspensions having 1:4 Fe_x_O_y_:water ratio were mixed with 100 mL of deionized water and 3 g of dried WB under stirring, and, finally, the mixtures were filtered the following day. Sorbents, before their application, were dried at laboratory conditions.

### 2.3. Methods

Particle size distribution measurements of the powder samples were performed using a Horiba laser scattering particle size distribution LA-950 (Horiba, Kyoto & Tokyo, Japan) in distilled water. Particle size analysis was performed with refractive index values of 1.500 for powder samples and 1.333 for distilled water.

The chemical composition of MWBI (0.8 g of FeSO_4_·7H_2_O) and MWBII (1.4 g FeSO_4_·7H_2_O) was determined using the SPECTRO XEPOS energy dispersive X-ray fluorescence spectrometer (EDXRF) (SPECTRO A. I., Kleve, Germany). The total content of Fe in these samples was determined after total decomposition in the mixture of acids using atomic emission spectroscopy with inductively coupled plasma (AES-ICP) SPECTRO ARCOSS (SPECTRO Analytical Instruments GmbH, Kleve, Germany). The Fe(II) was determined following the Czech standard [43].

The scanning electron microscope (SEM) images of WB and magnetically modified WB were obtained using the JEOL JSM-7610F Plus (JEOL Ltd., Tokyo, Japan) with EDS microanalyzer Aztec Line Standard Microanalysis system with Ultim Max 65 Analytical Silicon Drift Detector (SDD) (Oxford Instruments, High Wycombe, UK). Sample images details were taken by a secondary electron detector in the bright field mode.

FT-IR spectra in the mid-infrared range (400–4000 cm^−1^) were recorded on the Nicolet 6700 FT-IR device (Thermo Fisher Scientific, Waltham, MA, USA). ATR technique with diamond crystal and 32 scans were used. Spectra were analyzed in the OMNIC (OMNIC 8) and Origin software (OriginPro 9.1). All measured samples were, before the analysis, dried for 3 h at 80 °C to eliminate the influence of moisture absorption.

### 2.4. Characterisation of Magnetic Sorbents

The particle size distributions of the initial samples showed broad curves with an identical value of *d*_43_ = 46.3 μm (mean diameter) and *d_m_* = 51.5 μm (mode diameter) for WB and MWBI. The MWBII showed very broad particle size distributions with three size fractions with mode diameters *d_m_* = 0.39 μm, *d_m_* = 13.25 μm, and *d_m_* = 174.62 μm.

The total amount of Fe was determined to be 9.56 wt.% in MWBI and 29.1 wt.% in MWBII. Determination of the type of Fe_x_O_y_ in the prepared composites is still complicated. The analysis of pure Fe_x_O_y_ NPs, prepared in the first step of the modification procedure (described in Section 2.2), revealed the presence of γ–Fe_2_O_3_ (maghemite) and Fe_3_O_4_ (magnetite) and was performed in our previous publication [44]. Subsequent chemical analyses proved that 6.61 wt.% in MWBI and 5.73 wt.% in MWBII were present as FeO.

### 2.5. Potentiometric Titration

The titration experiment was carried out using titration under a nitrogen atmosphere to remove CO_2_. A total of 0.3 g of sample was added to deionized water with 5 mL of background electrolyte 0.01 M NaCl. The suspension was titrated to pH 3 using 0.1 M HCl, and then the suspension was titrated using the known volume of 0.1 M NaOH until pH 11 was reached. The surface charge was calculated based on the suspension electro-neutrality [45].

### 2.6. Sorption Experiments

Beryllium standard solution (Be_4_O(C_2_H_3_O_2_)_6_ in HNO₃ 0.5 mol/L 1000 mg/L Be Certipur^®^, MERCK, Darmstadt, Germany) was used for the preparation of solution needed for the optimization of time and pH parameters. Batch sorption experiments were carried out in plastic flasks at laboratory temperature (24 ± 1.5 °C). The effect of contact time on the sorption capacity of WB, MWBI, MWBII was studied in the range 0.5–48 h. The 50 mL of solution containing 1 mg/L of Be (pH 4) was shaken at a constant speed (28 rpm) with a 0.1 g (±0.01) sorbent dose. After shaking, a magnet was applied to the plastic flask, the liquid phase was separated, and before the AES-ICP analysis, it was filtered through 0.23 µm pore filter (Pragopor 8, PRAGOCHEMA spol. s.r.o., Praha, Czech Republic).

The pH value of the sorbate solution is one of the main factors affecting the sorption process. To study the pH effect, aqueous solutions containing Be (2 mg/L) were adjusted by 0.1 M HNO_3_ or 0.1 M NaOH solution in the interval 2–11. 0.1 g of WB, MWBI, and MWBII were mixed with 50 mL of the solution and shaken for 6 h.

To determine the maximum sorption capacity, the 50 mL of solutions containing 5–2000 µg/L of Be with initial pH 6 ± 0.5 were mixed with 0.1 ± 0.01 g of WB, MWBI, MWBII and shaken for 6 h. 

The equilibrium metal uptake capacity was calculated according to Equation (1):(1)$$q_{e}=\frac{V\times \left(c_{0}-c_{e}\right)}{m}$$
where *q_e_* (mg/g) is the amount of contaminant removed from the solution, *c*_0_ and *c_e_* are the concentrations of the metal ions in the initial solution and at the equilibrium after the sorption experiment (mg/L), *V* is the volume of the solution (L) and *m* is the sorbent dose (g) in the mixture.

To investigate the sorption mechanism, sorption data were tested by different kinetic models. The pseudo-first-order model (PFO) suggested by Lagergren [46] is described as:(2)$$\log\left(q_{e}-q_{t}\right)=\log\left(q_{teor}\right)-\frac{k_{1}\times t}{2.303}$$
where *q_t_* (mg/g) is the amount of Be(II) sorbed on the sorbent at the time *t* (min), *k*_1_ (1/min) is the pseudo-first-order (PFO) rate constant. *q_teor_* (mg/g) symbolizes the maximum calculated uptake of the pseudo-first-order model at the equilibrium [14].

Pseudo-second-order (PSO) kinetic model was suggested by Ho and McKay [47]:(3)$$\frac{1}{q_{t}}=\frac{1}{k_{2}\times q_{teor}^{2}}-\frac{t}{q_{teor}}$$
where *k*_2_ (g/(mg∙min)) is the pseudo-second-order rate constant, *q_teor_* (mg/g) is the maximum calculated uptake of the pseudo-second-order model at equilibrium.

The initial sorption behavior can be expressed by the Weber Morris model [48] written as:(4)$$q_{t}=K_{i}\times t^{1/2}+C$$
where *K_i_* (mg/(g∙min^0.5^)) is the intraparticle diffusion rate constant, and *C* (mg/g) is the intercept providing information on the thickness of the boundary layer of sorbent.

The initial sorption factor *R_i_* determined from the so-called characteristic curve based on the intraparticle diffusion (IPD) model can be expressed as:(5)$$R_{i}=\frac{q_{ref}-C}{q_{ref}}=1-\frac{C}{q_{ref}}$$
where *q_ref_* (mg/g) is the solid phase concentration in the longest time of sorption process [49]. 

The experimental data were analyzed using various isotherm models (Freundlich, Langmuir, Sips, and Redlich-Peterson [50,51,52,53,54,55,56,57,58], which are described in more detail in the Appendix A), providing important information about the sorption processes. The applicability of the isothermal model is mainly determined by comparing the correlation coefficients *R*^2^ [59].

### 2.7. Sorbents Regeneration

The sorbents renewability was tested by a sorption-desorption experiment to determine the sorbent recycling in a discontinuous rotating container for 24 h. The solid-to-liquid ratio (0.1 g of MWBI to 50 mL of extraction agent) was preserved. 0.01 M H_2_SO_4_ (denotes as extraction agent I), Na_2_CO_3_ (II), CaCO_3_ (with adding HNO_3_) (III), 0.01 M HCl (IV), and deionized water (V) solutions were used for the desorption of Be(II) from MWBI. The concentration of the desorbed Be(II) was determined by AES-ICP after filtration. The composite was dried at laboratory temperature, weighted, and the following sorption experiment was carried out. The efficiency of Be(II) removal was calculated as follows:(6)$$\mathrm{Be}\left(\%\right)=\frac{c_{des}}{c_{ads}}\times 100$$
where *c_des_* (mg/L) is the concentration of Be(II) determined in the extraction agent after the desorption experiment and *c_ads_* (mg/L) is sorbed Be(II) before the desorption experiment.

## 3. Results and Discussion

### 3.1. Characterisation of WB, MWBI, and MWBII

Total carbon content was determined to be 73.64 wt.% in WB and 4% of volatile compounds. The contents of the minor elements are present in Table 1. The total content of Fe in original WB, MWBI, and MWBII was determined after decomposition in a mixture of acid. After the magnetic modification of WB, the total amount of Fe increased from 4.82 wt.% to 9.56 wt.% in MWBI and 29.1 wt.% in MWBII.

#### 3.1.1. SEM-EDX Analysis

The SEM images of MWBI (Figure 1b) reveal clusters of particles smaller than in the original WB (Figure 1a), this corresponds with the magnetic modification of the WB surface, the surface detail of MWBII (Figure 1c) show more round particles and smoother surface particles than WB and MWBI. The SEM images details of MWBI and MWBII show that with increasing Fe content, the refinement of the particles increases as well. The Fe content, determined by SEM-EDX analysis, was lower in MWBI than in WB. These differences were caused by analysis of a smaller part of each sample and thus led to the observed difference in the Fe content between the samples. The EDX spectra show the lowest Fe content in the MWBI sample but, after recalculation, increasing Fe concentration was observed as follows: WB < MWBI < MWBII (Table 2). These differences were probably caused during the recalculation and could be influenced by the peak of the highest intensity of MWBI present in the EDX spectrum. This peak corresponded to carbon present in the samples and was included in the analysis, which influenced the determined chemical content. Moreover, the EDX analysis is not as accurate as XRPD analysis or chemical decomposition for the determination of chemical content; therefore, the differences were observed. The EDX spectra show that the magnetic modification of WB caused the decrease of minor elements content which probably leached from WB during the preparation of MWBI and MWBII (Table 1). A similar trend was observed in other elements, with the exception of potassium. Table 2 shows the chemical composition of WB, MWBI, and MWBII determined EDX analysis, while the chemical composition determined by XRFS is presented in Table 1.

#### 3.1.2. IR Spectroscopy

All studied samples were analyzed using FT-IR spectroscopy (see Figure 2) to determine the functional groups in the biochar because they play a key role in the metal sorption. After the pyrolysis, a significant loss of organic matter occurred, while the preserved organic matter was low on H content because no bands in 2950–2850 cm^−1^ were visible. The major polymeric components of biomass are hemicellulose, cellulose, and lignin, with a decomposition temperature of about 300 °C (cellulose and hemicellulose) and about 600 °C (most volatile organic compounds and lignin) [60,61,62]. Also, it is evident that the visible bands have very low intensity and broad character for the heterogeneous biochar and high content of amorphous materials, respectively [63]. In the WB spectrum, the bands of O–H vibration are missing, but a very broad band at ~3100 cm^−1^ is visible in the case of MBWI, which is more intensive in the case of MBWII and corresponds to the O–H stretching vibration. Since all samples were dried under the same conditions prior to the FT-IR measurements and measured directly after drying, this band is most likely not affected by air moisture and is related to the Fe content (its intensity increasing with the increase of Fe). The higher intensity of this band (spectrum MBWII) is also connected with the appearance of the weak band at 1628 cm^−1^, which belongs to the deformation vibration of O–H [64]. Other trends connected with the Fe content increase were observed in the spectra. A band belonging to the C=H deformation vibration at 1389 cm^−1^ (WB) [65] was shifted to the 1411 cm^−1,^ and its intensity decreased in MBWI and vanished in the case of MBWII. Similarly, the same trend can be observed for bands at 1017 and 869 cm^−1^ (WB) which were shifted to 1019 and 871 cm^−1^ (MBWI) and missing (or only as very weak shoulder present) in the spectrum MBWII. These bands could be associated with vibrations of Si–O–Si or—more probably due to small content of SiO_2_—to C–O and or C–H deformation vibrations of aromatic rings, respectively. Moreover, new bands appeared in the spectrum MBWII: Shoulder appearing at 660 cm^−1^ can be connected with the presence of the Fe–O stretching vibration
[66] and band at 1129 cm^−1^ which could be attributed to C–O vibration [67].

### 3.2. Potentiometric Titration

The potentiometric titrations were performed to determine the negative charge of the sorbents since electrostatic interactions between negatively charged surface and positively charged ions could serve as a binding mechanism. Figure 3 presents the surface charge of original WB, MWBI, and MWBII. The surface charge of sorbent with the highest Fe_x_O_y_ content corresponds almost totally with the course of the non-modified WB curve. The MWBI have a slightly higher negative charge than WB and MWBII, with significant charge at alkaline pH.

The zero point of charge (pH_zpc_) of WB, MWBI, and MWBII was found to be 5.38, 4.17, and 5.85, respectively. These results correspond well with Figure 4, where the sorbent amount of Be(II) was significantly increased at the pH 4.95 for MWBI, 5.85 for MWBII, and original WB. Below the pH_zpc_, the sorbents were positively charged and, therefore, unfavorable for Be(II) sorption. However, with increasing pH, the negative surface charges of sorbents increased, which led to an increase in the Be(II) sorption [68].

The titration curves show a slight decline around pH 6 and a rapid decline around pH 10, which is a similar trend observed by Mai et al. [69]. This predicts the presence of functional groups around the mentioned pH values buffering the solution [45]. These groups could be carboxyl (pK_a_~3 to 5) and hydroxyl groups (pK_a_~10) [70], which corresponds with their presence in FT-IR spectra (Figure 2).

### 3.3. Kinetic Test and the Effect of pH

#### 3.3.1. Effect of pH

The beryllium removal from the aqueous solution increased with increasing pH (Figure 4). At low pH (2.00–3.00), the sorption of metal cations is unfavorable and restricted. The maximum sorption capacity was less than 1.6 mg/g. At low pH, the active sites (carboxyl and hydroxyl groups) of sorbents are protonated; therefore, the repulsive forces are responsible for unfavorable metal sorption [71]. In the 3–5 pH interval, the sorption capacity increased with increasing pH and reached 2.4 mg/g for both MWBII and original WB. The active sites are deprotonated and allow metal cations binding. Significantly increased Be(II) sorption was achieved in the 5–9 pH interval. The maximum sorption capacity was achieved at pH = 6, which, however, still did not prevent the precipitation of Be(II) in the aqueous solution. Be occurs mainly in the form of Be^2+^ and Be(OH)^+^ at pH 4–6, and precipitation of Be to Be(OH)_2_ starts when pH exceeds 6 [68]. Beryllium is expected to exist as insoluble Be(OH)_2_ in solutions with a pH of 7 to 12 when its concentration is 1 mg/L. Moreover, based on the results of Boschi and Willenbring [72] at pH 7, most of Be(OH)_2_ can be removed from the solution via filtration with a 0.45 μm filter. In the present experiment, the Be(II) started to precipitate in solutions with a pH of 10 and 11 because the initial concentration of Be(II) in the initial solution (determined by AES-ICP) decreased to 0.748 mg/L (for pH 10) and 0.545 mg/L (for pH 11). Therefore, the sorption amount decreased significantly—due to the precipitation of Be(II) in solutions and, therefore, low initial Be(II) concentration—while the Be(II) removal was unchanged.

#### 3.3.2. Kinetic Parameters

A higher amount of Be(II) was adsorbed onto MWBI (0.57 mg/g) than MWBII (0.38 mg/g). The maximum equilibrium capacity was achieved in 6 h for MWBI and 48 h for MWBII (Figure 5a,b). These differences may be caused by a higher content of Fe_x_O_y_ adsorbed onto MWBII due to the Fe_x_O_y_ occupation of the active sites of WB. The efficiency of Be(II) removal was higher than 90% for MWBI and 80% for MWBII after 6 and 48 h, respectively. 

The kinetic study determined the equilibration time for sorption and revealed the reaction rate. The PFO and PSO kinetic parameters were calculated using Equations (2) and (3) determined from Figure 6 and are presented in Table 3. The PSO kinetic model (*R*^2^ > 0.95) fitted experimental sorption data of Be(II) onto MWBI and MWBII better than PFO kinetic model (*R*^2^ > 0.76). The kinetic rate constants predict the fastest sorption of Be(II) onto MWBI than MWBII, which corresponds with the data presented in Figure 5. The calculated *q_teor_* values were consistent with the experimental results, besides *q_teor_* determined for MWBI by PFO. The high amount of Be(II) removed from solution by MWBI during the initial time can probably be explained by a higher number of binding sites on the MWBI surface. The MWBII removed 85% of Be(II) from the solution after 48 h of the contact time, which means that with the increasing sorption time, the removal of Be(II) from the solution increases too, albeit after 24 h of contact time slowly—the removal efficiency of sorbent increased by only 10%. The further increases in contact time probably did not enhance the sorption efficiency significantly. These differences were caused by the difference in the Fe_x_O_y_ content in both sorbents. The surface and active sites of MWBII were probably more saturated than the MWBI surface by Fe_x_O_y_; therefore, the sorption rate was faster, and the adsorbed amount of Be(II) was, during the 48 h of contact time, higher in MWBI than MWBII. Various sorption rates could result from the structural changes—as evidenced by the slight decrease of band intensity at 869 cm^−1^—observed in the FT-IR spectrum. Therefore, there is a presumption, which was confirmed in part 3.4. Sorption test that MWBII could provide more binding sites due to the presence of Fe_x_O_y_ on its surface and, we assume, is a more effective sorbent than MWBI.

The values of intercept (C)—providing an idea about the thickness of the boundary layer—determined for Be(II) sorption onto MWBI is greater than values determined for MWBII [73]. Therefore, the effect of the boundary layer effect is greater in the case of Be(II) sorption onto MWBI than onto MWBII.

The *R_i_* values characterizing the Be(II) sorption onto MWBI and MWBII determined by the IPD model are presented in Table 3. Strong initial sorption shows sorption Be(II) onto MWBI, while weak initial sorption was determined in the case of MWBII.

Figure 7 shows the plot of Be(II) uptake by MWBI and MWBII versus square root of time (sorption duration). It is apparent that the plots are not linear over the time range. If the intraparticle diffusion is the only rate-limiting step, the plot passes through the origin. If not, the sorption is affected to some degree by the boundary layer. As can be seen in Figure 7, the first slope belonging to MWBI, indicated by the sorption rates, does not pass through the origin; therefore, the sorption was limited by the boundary layer. The second part can be attributed to the diffusion into macropores, and the third part refers to the diffusion into the micropores and final equilibrium stage for which the intraparticle diffusion starts to slow down due to the low adsorbate concentration left in the solution. Therefore, the limiting step of Be(II) sorption is the diffusion into MWBI pores [74,75]. The course of the slope characterizing the sorption of Be(II) onto MWBII is nearing the beginning; the second part of the slope part is attributed to the diffusion into macropores and wider mesopores. Therefore, the intraparticle diffusion can be considered the rate-limiting step.

### 3.4. Sorption Test

Based on *R*^2^ values determined by interpolation of experimental data for MWBI and MWBII (Figure 8), there were no significant differences between the sorption isotherms (Table 4). The conclusions are influenced by low values of correlation coefficients (*R*^2^). To describe the Be(II) sorption onto MWBI, none of the isotherms could be used—values of correlation coefficients were lower than 0.31. With the increasing initial concentration, the *q_e_* values increased slowly, and the maximum sorption capacity—1.44 mg/g—was achieved for MWBI. The Langmuir adsorption isotherm predicts that 4.31 mg/g of Be(II) could be sorbed onto MWBII. With the increasing initial concentration, the experimental *q_e_* values increased and were determined to be 1.45 mg/g of Be(II) onto MWBII. Based on the course of the experimental data curve, we can predict that MWBII could be better sorbent than MWBI due to the presence Fe_x_O_y_ and, therefore, more active sites. Both composites seem to be suitable for the final cleaning aqueous solutions.

The values of *R_L_* at pH 6 (Table 4) were found to be 0 ≤ *R_L_* ≤ 1, which indicates the favorable sorption of Be(II) onto MWBI and MWBII.

The marginally higher adsorbed amount of Be(II) was observed onto MWBII with higher content of Fe_x_O_y_ than in the case of MWBI sorbent, which was consistent with results presented by [76,77]. This effect may be related to the presence of an intensive band of O–H in FT-IR spectrum of MWBII (Figure 2). The comparison of various types of Be sorbents and their maximum adsorbed amount is presented in Table 5. The carboxyl and hydroxyl groups were determined for most sorbents as the active sites, and the optimal pH value for the Be(II) sorption was found in the pH interval 4–7. The studies dealing with sorption of Be(II) are scarce; however, study focused on the sorption of Be(II) by either biochar or magnetically modified biochar has been missing so far.

### 3.5. Desorption Test

The desorption of Be(II) was affected mainly by acidic extraction agents (I, III, IV)—more than 95% of sorbed Be(II) was removed (Figure 9). The Be(II) reacted in the acidic environment forming beryllium salts and hydrogen, therefore was easily desorbed from the surface than by extraction agent V. Less than 40% of Be(II) was desorbed by the extraction agent II and 2% by the extraction agent V. Due to the low amount of Be(II) desorbed to the extraction agent V, the following sorption was not carried out. It can be concluded that the manipulation with composite in the aquatic environment after sorption is—from the point of its stability—safe. However, the following sorption experiment revealed that the acidic solutions probably disrupted the active sites of MWBI; therefore, the Be(II) sorption was lower than 6% (after the desorption experiment in the extraction agents I, III, IV). The MWBI and MWBII magnetic properties were verified using a magnet after sorption/desorption experiments—both composites retained their magnetic properties.

## 4. Conclusions

Pyrolysed waste material was used for the preparation of two composites with magnetic properties. The composites with various Fe_x_O_y_ content were prepared using a two-step method. The results of the kinetic study show that Be(II) was eliminated faster by MWBI than by the MWBII composite in the first minutes; however, the Langmuir sorption model predicts higher sorption capacity for MWBII (4.31 mg/g) than for MWBI. The desorption test also confirms good affinity of Be(II) to sorbent; therefore, it can be concluded that the manipulation with composite is safe. The Fe concentration determined after the sorption experiment was lower than 0.02 mg/L in a rather broad range of pH values (pH interval 4–11); therefore, the composites were stable. The results confirmed that both the production of biochar from poor-quality wheat and its magnetic modification are suitable for application in the final cleaning process of water treatment; moreover, the composites can be easily removed using a magnet.

## Figures and Tables

**Figure 1 materials-14-06610-f001:**
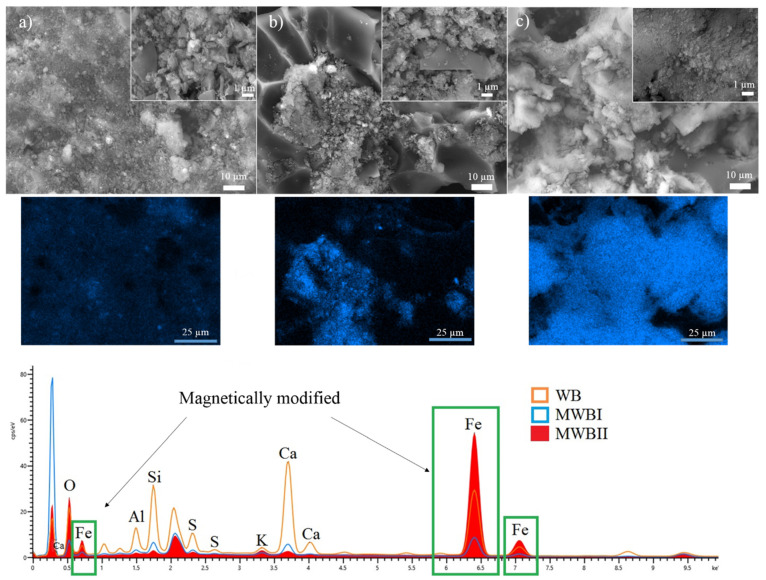
SEM-EDX analysis, images details, and mapping of Fe on the samples (**a**) WB, (**b**) MWBI, and (**c**) MWBII.

**Figure 2 materials-14-06610-f002:**
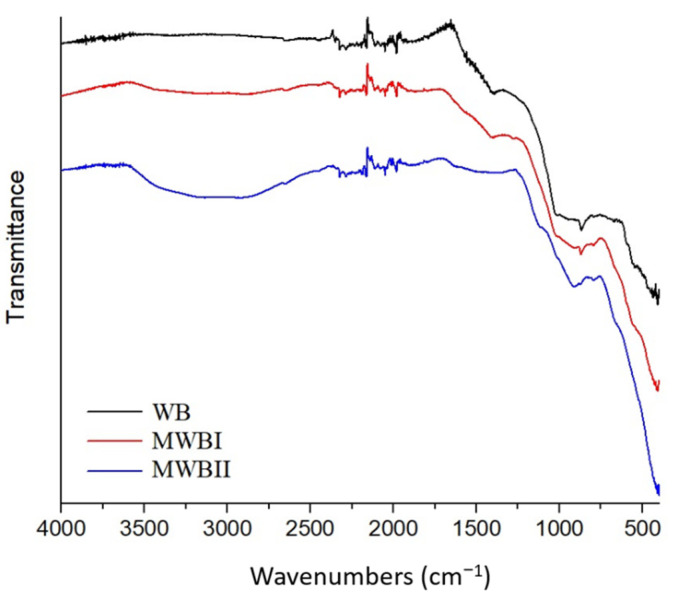
The FT-IR spectrum of WB, MWBI, and MWBII.

**Figure 3 materials-14-06610-f003:**
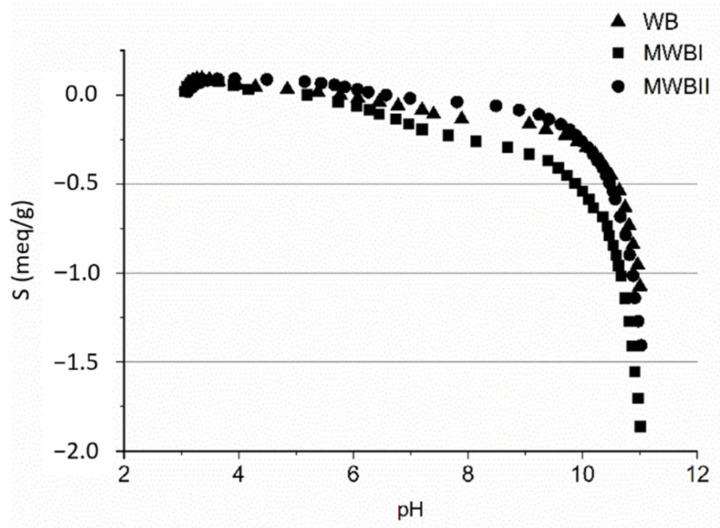
Potentiometric titration curve of untreated WB, magnetically modified MWBI, and MWBII at a background electrolyte NaCl.

**Figure 4 materials-14-06610-f004:**
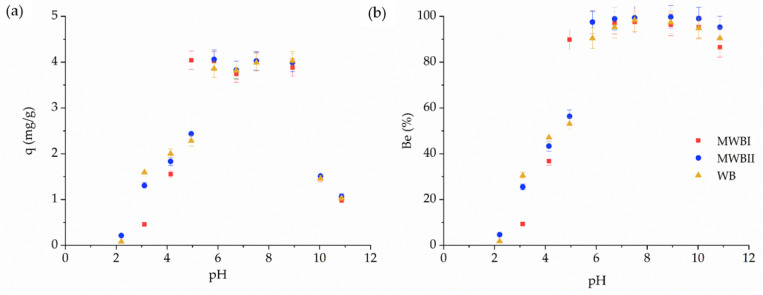
Effect of pH on Be sorption from aqueous solution using the MWBI, MWBII and origin WB (2 mg/L Be; 6 h contact time, 0.1 g:50 mL sorbent dosage, 25 °C); (**a**) sorption capacity (in mg/g) and (**b**) removed amount (in %).

**Figure 5 materials-14-06610-f005:**
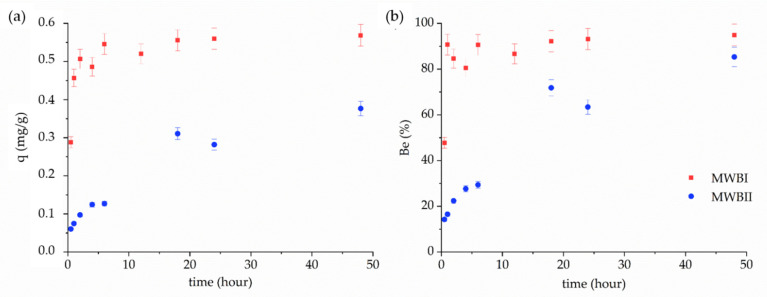
Kinetics (**a**) and efficiency (**b**) of Be(II) removal from aqueous solution by MWBI and MWBII; pH 4, 0.1:0.05 solid-to-liquid ratio.

**Figure 6 materials-14-06610-f006:**
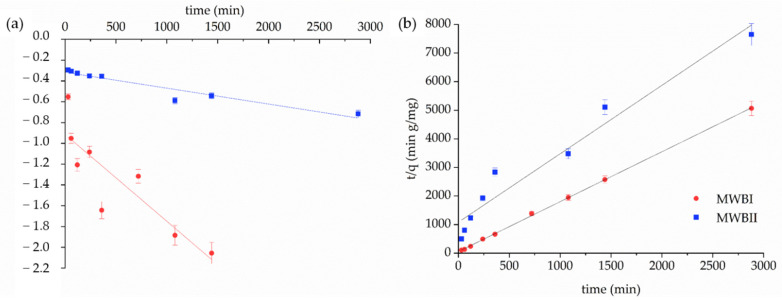
Sorption rate of Be(II) ions on the MWBI and MWBII calculated by (**a**) the PFO and (**b**) the PSO kinetic models.

**Figure 7 materials-14-06610-f007:**
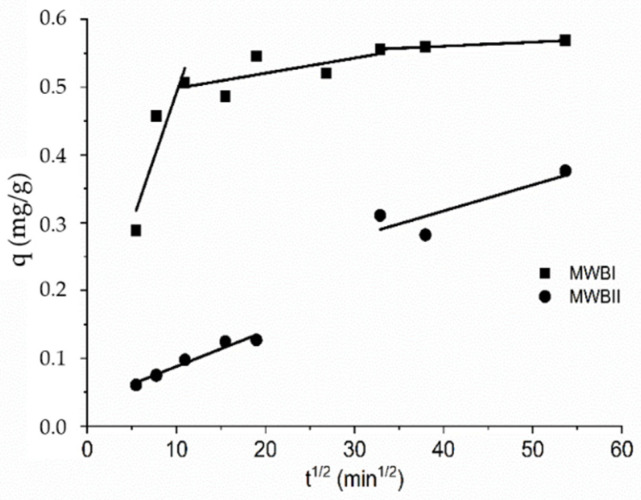
The Weber Morris model for sorption of Be(II) onto MWBI and MWBII from aqueous solution.

**Figure 8 materials-14-06610-f008:**
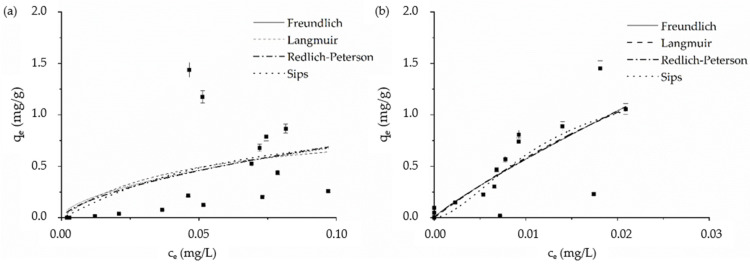
The fitted experimental data obtained from sorption Be(II) onto (**a**) MWBI and (**b**) MWBII by Freundlich, Langmuir, Redlich-Peterson, and Sips isotherm model.

**Figure 9 materials-14-06610-f009:**
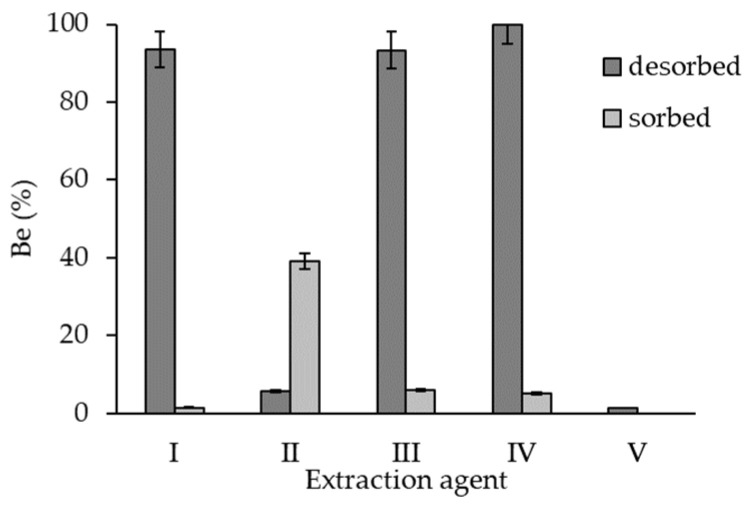
The efficiency of Be(II) desorbed from MWBI by extraction agent: I—0.01 M H_2_SO_4_, II—Na_2_CO_3_, III—CaCO_3_ with added HCl, IV—0.01 M HCl solution, V—deionized water; and efficiency of subsequent sorption of 1 mg/L Be(II).

**Table 1 materials-14-06610-t001:** The total content of minor elements determined in WB, MWBI, and MWBII expressed in oxides, besides the Fe and Cl. LOI—lost on ignition.

Content	WB (wt.%)	MWBI (wt.%)	MWBII (wt.%)
Fe	4.82	9.56	29.1
SiO_2_	3.56	3.65	2.89
K_2_O	0.88	1.36	0.80
CaO	3.99	2.37	1.32
Na_2_O	0.77	0.64	0.59
MgO	0.56	0.58	0.26
Al_2_O_3_	0.68	0.54	0.33
P_2_O_5_	3.61	2.97	1.54
SO_3_	0.59	1.04	0.76
Cl	0.003	0.04	0.03
TiO_2_	0.15	0.07	0.05
Cr_2_O_3_	0.10	0.06	0.04
MnO	0.10	0.07	0.08
ZnO	0.81	0.49	0.40
LOI	79.84	74.57	57.58

**Table 2 materials-14-06610-t002:** The chemical composition of WB, MWBI, and MWBII in atomic % determined by EDX analysis.

Chemical Composition	WB	MWBI	MWBII
O	70.05	74.63	68.44
Al	3.23	2.32	0.55
Si	6.96	5.18	0.86
S	1.94	1.73	0.38
K	0.43	0.88	0.54
Ca	8.03	3.97	0.44
Fe	9.37	11.30	28.80
Total	100.00	100.00	100.00

**Table 3 materials-14-06610-t003:** Kinetic parameters of the PSO kinetic model determined by linear regression.

Kinetic Model	Model Parameters	MWBI	MWBII
Pseudo-first-order	*q_teor_* (mg/g)	0.122	0.482
	*q_e_* (mg/g)	0.569	0.377
	*k*_1_ (1/min)	0.002	0.0005
	*R* ^2^	0.761	0.917
Pseudo-second-order	*q_teor_* (mg/g)	0.573	0.418
	*k*_2_ (g/(mg.min))	0.057	0.130
	*R* ^2^	0.9996	0.953
Weber Morris model	*K_i_* (mg/(g.min^0.5^))	0.0022	0.0069
	*C* (mg/g)	0.4698	0.034
	*R* ^2^	0.709	0.959
IPD model	*R_i_*	0.174	0.912
		Strongly initial sorption	Weakly initial sorption

**Table 4 materials-14-06610-t004:** Sorption parameters determined by Langmuir, Freundlich, Redlich-Peterson, and Sips isotherm models in non-linear form.

Kinetic Model	Isotherm Parameters	MWBI	MWBII
Freundlich	*K_F_* (mg/g)	2.57	30.8
	*n*	1.74	1.15
	*R* ^2^	0.226	0.625
Langmuir	*q_m_* (mg/g)	0.944	4.31
	*K_L_* (L/mg)	21.4	15.6
	*R* ^2^	0.250	0.628
	*R_L_*	0.024	0.032
Redlich-Peterson	A	74.2	266
	B	26.1	7.89
	g	0.43	0.166
	*R* ^2^	0.227	0.626
Sips	*q_m_* (mg/g)	0.892	1.47
	*K_S_* (L/g)	4372	4348
	*n* (L/mg)	0.009	0.03
	*R* ^2^	0.305	0.667

**Table 5 materials-14-06610-t005:** The comparison of various types of sorbents for Be(II) removal.

Sorbent	Sorbent Dose	pH	Time (Min)	Active Sites	*q_exp_* (mg/g)	Ref.
Sulfated cross-linked chitosan	20 mg/50 mL	4	100	Hydroxyl, sulfonic, amino	40.6	[14]
Polystyrene-azo-3,4-dihydroxybenzene	10 mg/20 mL	3.2–6.1	30	Hydroxyl	22.5	[15]
Mesoporous silica 5-nitro-2-furaldehyde synthesized	10 mg/10 mL	7.2	30	-	0.034	[78]
Aerobic granule	3 g/1 L	6	300	Carboxyl, hydroxyl	14	[68]
Polystyrene Based Chelating Polymer	10 mg/20 mL	2.1–6.1	30–90	-	3.7–22.5	[15]
Amberlite IR-120	0.8 g/100 mL	3.5	120	-	11.7–38.7	[16]
Magnetically modified biochar	0.1 g/50 mL	6	360	Hydroxyl, carboxyl	1.452	Present study

## Data Availability

Data is contained within the article.

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
