# Peer review of "Magnetically Modified Biosorbent for Rapid Beryllium Elimination from the Aqueous Environment"

_materials, 2021, doi:10.3390/ma14216610_

Round 1
Reviewer 1 Report
The obtained results may actually be helpful in developing a method of obtaining an efficient sorbent from natural sources, especially since the wheat used is not suitable for any other application. Therefore, it is not a material that could be used, for example, as animal feed. The authors chose the methodology well. However, some clarifications and additions are missing. English should also be corrected.
Please, here are some of my comments:
- p.2, l.86: What was the pyrolysis time?
- p.2, l.88: What was the size of the ground particles?
- p.2, l.92-93: The description of the substrates should be in the sub-section: Materials.
- p.3, l.104: How were the sorbent samples dried after they were received? Were the magnetic properties of both sorbents determined and how?
- p.6, Table 1: What does it means LOI?
- p.6, Fig.1: The amount of Fe in MWBI is smaller than in WB?
- p.7, l.256: In fact, the removal of beryllium from the aqueous solution increased with increasing pH, however, at around 10-11, it decreased drastically. Why?
- p.8., l.256: Describe which functional groups are these active sorption centres.
- p.8. Fig.4: What is the difference between Fig. 3 and Fig. 4?
- p.10, Fig.6: Did you check q in the initial time (0-5 min1/2)?
- p.10, Fig.6: Have you checked the time after which equilibrium sorption occurs for MWBII?
- p.10, Fig.7: For MWBII it can be seen that not all sorption centres have been occupied (the Langmuir isotherm has not "reached" the maximum occupancy yet). For what reason?
- p.11, l.340: How was the magnetic properties of sorbent tested?
- p.11, Table 4: The way of writing active sites should be standardized - they should be given either by a chemical formula or by a name in all Table.
- p.12, Table 4: What was the concentration of Be(II) in each sample? You used a much smaller amount of adsorbent than in the examples of other research groups. It's hard to compare the results.
- p.13, Figure 9: In the case of Na2CO3, the desorption is small, but the sorption is much higher than in other cases. This indicates the possibility of reusing this sorbent after desorption in Na2CO3. Have you checked the next desorption-sorption cycle? how many times could such a sorbent be used?
- p.13, l.401-402: Does this conclusion follow from this manuscript? I do not see where you describe the methodology and conclusions from the removal of Be(II) by the magnetic way.
Author Response
Thank you for your time you devoted to revised our manuscript. We carefully answered all questions and highlighted all changes are in the manuscript in yellow. English was checked as well. Please see the attachment.

Reviewer 2 Report
Journal: Materials
Title: Magnetically modified biosorbent for rapid beryllium elimination from the aqueous environment
This is a classic work in the field of sorption, sorption of Be from aqueous solutions on 3 biosorbents, WB, MW BI and MWB II. The paper seemed promising based on the Introductory Part and the Materials and Methods. However, the results and discussion are very confused and unsystematic explained, the Figure of sorption kinetics is missing, the conclusions are inadequate. Therefore, a complete reorganization of the paper is needed, better explanations, thereafter the paper can be considered for a possible publication.
Therefore, I suggest a major revision.
My comments follow:
- Recommendation only: use of the word elimination, elimination efficiency vs. removal, removal efficiency
- Sorbent vs adsorbent; adsorption or sorption capacity - uniform throughout the manuscript
- ml, L vs.mL, L or ml, l - uniform throughout the manuscript
INTRODUCTION
Line 26: “(NTP 2016)” - is this cited literature or?
Line 81-82: “The magnetic modification of biochar originated from poor-quality wheat has not been tested yet.” - I think something should be added in this sense: as far as we know to date based on the available literature.
MATERIALS AND METHODS
Lines 85-90: specify the pyrolysis time
Line 92: In the chemical formula use a multiplication sign, not a dot, in all parts of the manuscript
Line 92: per analyses or pro analyses
Line 96: “The magnetic waste biochar (MWB) – with different Fe amount” - I would rather write: The magnetic waste biochar (MWB) – with 0.8 g and 1.4 g of FeSO4…..or Two magnetic waste biochar (MWB)….
Line 107: one parenthesis is redundant
Lines 120-121: I would emphasize: before the analysis
Lines 130-136: State the volume of Be solution
Line 157: one parenthesis is redundant
Lines 159-160: I think this is a Weber Morris model; In Table 2, however, it is listed as the W-M model
Note: the order of the description of the experiments MUST follow the order of the explanation of the results in Chapter 3
RESULTS AND DISCUSSION
Line 208: arepresent to are present
Lines 210-11: “After the magnetic modification of WB, the total amount of Fe increased from 4.82 wt.% to 9.56 wt. % in MWBI and 29.1 wt.% in MWBII.” - Compare these results with the results in Table 1, I mean the Fe content.
Line 214 “clusters of particles” and line 215 “finer particles” - Define the chemical composition of the particles, which clusters are in question
Line 252: Section title 3.2. is inadequate because it does not only explain the kinetic results
3.2.1. Effect of pH
Clearly define the optimal pH in one sentence. Do not use the symbol “Re” on the ordinate, which you did not explain before, the suggestion: removal efficiency,
3.2.2. Kinetic parameters
An adequate picture of the kinetic results is missing!
Compare all qe values for 3 biosorbents.
The whole section needs to be better explained.
In the title of Figure 4 you mention pH = 6, in Materials and Methods you state pH = 4 for the kinetic experiment (lines134-135)????????
Lines 287-288: “This was caused by the difference in the FexOy content in both sorbents. The surface and thus – probably – active sites of MWBII were more saturated than MWBI surface by FexOy;”- Are you sure? Usually the increase of Fe causes better sorption due to the presence of Fe-oxyhydroxide on the surface of the sorbent, i.e. Fe species act as active sites.
Furthermore, is there a release of Fe from the structure of MWBI and II biosorbents or its leaching? Did you measure it?
Also in lines 111-112 you stated: “Determination of the type of FexOy is described in more
detail in Tokarčíková et al. (2017) [44].” - I think it is necessary to look back briefly and explain again in the manuscript which types of Fe are involved. This will help clarify why sorption with a higher amount of Fe used in the modification process is not more efficient. A little help, what is the optimal dose of Fe?
Line 295: “scharacterising” ??
Lines 295-301: The W-M model should be explained first, followed by the IPD
The title of Figure 6 is not correct, it is a W-M model, not an IPD
Why were the results of the WB sample in Table 2 and Figure 6 not taken into account?
3.3. Adsorption test
Starting a chapter with a Figure without text and referring to the Figure that follows is inadmissible!
The results of the WB sample in Figures 7 a and b are missing.
I think testing the results according to sorption models is unnecessary because all used model does not fit experimental results. Therefore, the conclusion is " Langmuir model seem to be the best model to describe the adsorption of Be(II) onto MWBI. " incorrect!
Based on these models nothing can be concluded. It is better to explain the influence of the initial concentration on the sorption of Be.
Lines 339-340: This sentence is not related to the previous paragraph neither the next!
Line 343: “The values of R L at pH 6” – pH=6??? - in line 143 is written pH = 5-6???
Lines 345-346_ “Marginally higher adsorbed amount of Be(II) was observed onto MWBII with higher content of FexOy than in the case of MWBI sorbent,” - this is in contrast to your kinetic results.
What is the purpose of Table 4? Explain and compare the data from Tab 4 textually.
3.4. Potentiometric titration
MWB1 and MWB2 vs. MWBI and MWBII
The part in lines 370-374 should be before the part in lines 364-369.
Furthermore, this chapter MUST be part of chapter 3.1., and explanations related to the connection with Figure 3 should be in chapter 3.2.1.
3.5. Desorption test
In section Materials and methods, you did not state that you have performed sorption after desorption!
CONCLUSION - completely modified
Finally, you should define the mechanism of Be sorption on sorbents based on kinetic and sorption models, and even better if you have SEM-EDS images of saturated samples. Also, sorption chemistry would be interesting. Connect the sorption mechanism with the desorption test.
Author Response
Thank you for your time you devoted to revised our manuscript. We carefully answered all the questions and highlighted all changes in the manuscript in yellow. English was checked as well. Please see the attachment.

Round 2
Reviewer 1 Report
Thanks to the authors for clarifying my doubts and for including my comments in the manuscript. I believe the paper is now complete and can be published. However, I strongly suggest some sorting out in terms of the Materials / Methods subsections. Please take a look at my remarks:
Please, look carefully if the description of the materials is in the Materials section, and the description of the methods in the Methods. For example, a method (EDXRF) (l.89) Should be described in the Methods section, not Materials. And the description of materials from l.92-94 was not transferred to Materials. And a few other mistakes.
The particle size of WB and MWBI (l.111) should be reported in the Materials section (if this is not new information) or in the Results section, if these values are new, obtained by you for the purpose of this study.
According to your explanation to point 6: The amount of Fe in MWBI is smaller than in WB?
Your answer: The Fe amount determined in MWBI (after decomposition of samples in the acid, please see the section 3.1. Characterisation of WB, MWBI and MWBII) is higher than in WB. However, the SEM-EDX analysis was performed only on small part of the samples, which lead to the observed difference. – this explanation should be added to the manuscript.
The same in case of explanation to point 7: In fact, the removal of beryllium from the aqueous solution increased with increasing pH, however, at around 10-11, it decreased drastically. Why?
Beryllium is expected to exist as insoluble Be(OH)2 in solutions with a pH of 11 and 12 (Sun, F.; Sun, W.-L.; Sun, H.-M.; Ni, J.-R. Biosorption behavior and mechanism of beryllium from aqueous solution by aerobic granule. Chem. Eng. J. 2011, 172, 783-791. https://doi.org/10.1016/j.cej.2011.06.062). Therefore, there is presumption that the Be(II) precipitated to insoluble Be(OH)2 which was removed from the solution during the filtration and was not adsorbed to the MWBI and MWBII. – this explanation also should be added to the manuscript.
Author Response
Thank you for your time you devoted to revised our manuscript. We carefully answered all questions and highlighted all changes are in the manuscript in grey. Please see the attachment.

Reviewer 2 Report
Line 4: The "a" sign does not need to be associated with names since all scientists come from the same institution
Previous comment:
Line 92: In the chemical formula use a multiplication sign, not a dot, in all parts of the manuscript
New comment: Not corrected! Now in line 91: FeSO4 .7H2O = FeSO4·7H2O
Previous comment:
Lines 159-160: I think this is a Weber Morris model; In Table 2, however, it is listed as the
W-M model
New comment: Not corrected! Now in line 172 change Weber ti Weber-Morris
Previous comment:
Lines 210-11: “After the magnetic modification of WB, the total amount of Fe increased from
4.82 wt.% to 9.56 wt. % in MWBI and 29.1 wt.% in MWBII.” - Compare these results with
the results in Table 1, I mean the Fe content.
Thank you for your remark. The chemical content of minor elements in Table 1 are presented in the form of oxides. Therefore, the amount of total Fe and amount of Fe2O3 is different.
New comment: This can be easily converted to Fe content
Line 157: “2000 µg/L of Be with initial pH 6 ± 0.5 were mixed with” - why did you use this pH if you stated in line 302 that above pH = 6 precipitation occurs.
Previous comment:
Line 214 “clusters of particles” and line 215 “finer particles” - Define the chemical
composition of the particles, which clusters are in question
Answer: The chemical composition of presented images is defined in the spectra of EDX analysis. And the composition of atomic % are shown in Table 1.
New comment: I am of the opinion that the results of the EDS analysis (Table in response) should be included in the manuscript and explained in more detail.
Previous comment:
3.2.1. Effect of pH
Clearly define the optimal pH in one sentence. Do not use the symbol “Re” on the ordinate,
which you did not explain before, the suggestion: removal efficiency,
Thank you for reminder. The “Re” was changed by Be(%) which was defined in Eq.
11.
New comment: It is not corrected either in the text or in the Figure.
Previous comment: Lines 339-340: This sentence is not related to the previous paragraph neither the next!
New comment: I meant of this sentence:
The MWBI and MWBII magnetic properties were verified by magnet after sorption/desorption experiments and both composite retained their magnetic properties.
Previous comment:
Line 343: “The values of R L at pH 6” – pH=6??? - in line 143 is written pH = 5-6???
The pH value was specified.
New comment: Is pH = 6 adequate, precipitation???
The conclusions have not been corrected
Author Response

(The authors gave the same response as above.)
